# Going over Fungal Allergy: *Alternaria alternata* and Its Allergens

**DOI:** 10.3390/jof9050582

**Published:** 2023-05-18

**Authors:** Eva Abel-Fernández, María José Martínez, Tania Galán, Fernando Pineda

**Affiliations:** Applied Science, Inmunotek S.L., Parque Científico Tecnológico Alcalá de Henares, 28805 Madrid, Spain; efernandez@inmunotek.com (E.A.-F.); mjmartinez@inmunotek.com (M.J.M.); tgalan@inmunotek.com (T.G.)

**Keywords:** fungal allergy, *Alternaria alternata*, molecular biology

## Abstract

Fungal allergy is the third most frequent cause of respiratory pathologies and the most related to a poor prognosis of asthma. The genera *Alternaria* and *Cladosporium* are the most frequently associated with allergic respiratory diseases, with *Alternaria* being the one with the highest prevalence of sensitization. *Alternaria alternata* is an outdoor fungus whose spores disseminate in warm and dry air, reaching peak levels in temperate summers. *Alternaria* can also be found in damp and insufficiently ventilated houses, causing what is known as sick building syndrome. Thus, exposure to fungal allergens can occur outdoors and indoors. However, not only spores but also fungal fragments contain detectable amounts of allergens and may function as aeroallergenic sources. Allergenic extracts of *Alternaria* hyphae and spores are still in use for the diagnosis and treatment of allergic diseases but are variable and insufficiently standardised, as they are often a random mixture of allergenic ingredients and casual impurities. Thus, diagnosis of fungal allergy has been difficult, and knowledge about new fungal allergens is stuck. The number of allergens described in Fungi remains almost constant while new allergens are being found in the Plantae and Animalia kingdoms. Given Alt a 1 is not the unique *Alternaria* allergen eliciting allergy symptoms, component-resolved diagnosis strategies should be applied to diagnose fungal allergy. To date, twelve *A. alternata* allergens are accepted in the WHO/IUIS Allergen Nomenclature Subcommittee, many of them are enzymes: Alt a 4 (disulfide isomerase), Alt a 6 (enolase), Alt a 8 (mannitol de-hydrogenase), Alt a 10 (aldehyde dehydrogenase), Alt a 13 (glutathione-S-transferase) and Alt a MnSOD (Mn superoxide dismutase), and others have structural and regulatory functions such as Alt a 5 and Alt a 12, Alt a 3, Alt a 7. The function of Alt a 1 and Alt a 9 remains unknown. Other four allergens are included in other medical databases (e.g., Allergome): Alt a NTF2, Alt a TCTP, and Alt a 70 kDa. Despite Alt a 1 being the *A. alternata* major allergen, other allergens, such as enolase, Alt a 6 or MnSOD, Alt a 14 have been suggested to be included in the diagnosis panel of fungal allergy.

## 1. Classification, Morphology and Distribution of *Alternaria alternata*

Fungi are heterotrophic eukaryotic organisms, and their reproduction can be sexual by spores (e.g., ascospores, basidiospores, zygospores), asexual by spores or mitospores (conidiospores and spores, created in sporangia), or by fragmentation of the mycelium, budding (in yeasts), and fission (vegetative reproduction).

Fungi are cosmopolitan organisms. They can be found in any region of the world. More than 100,000 species have been described so far. Nevertheless, it is estimated that there are at least one million and a half species [1]. Of these, only a few hundred have been described as opportunistic pathogens, producing disease in humans through three specific mechanisms: direct infection, induction of dysregulated immune responses and secondary metabolite toxicity. In general, exposure to fungi occurs by inhalation, skin contact or ingestion, although the inhalation route is the most important in terms of producing respiratory symptomatology.

Among the pathogenic fungal species described, about 80 genera produce type I allergy (IgE antibody-mediated), which usually manifests as allergic rhinitis and rhinosinusitis, allergic asthma and atopic dermatitis [2]. The most relevant allergenic fungi belong to the phylum Ascomycota, followed by the phyla Basidiomycota and Zygomycota. The genera *Alternaria, Cladosporium*, *Aspergillus* and *Penicillium*, belonging to the phylum Ascomycota, are considered the most relevant allergenic sources [2,3,4]. Other less relevant allergenic fungi are *Candida, Fusarium* or *Curvularia* (Ascomycota), *Malassezia* (Basidiomycota) or *Rhizopus* (Zygomycota), among others [2]. *Alternaria alternata* has a worldwide distribution and a high presence in the environment. The *Alternaria* spores are considered one of the most abundant and potent sources of sensitising airborne allergens [5].

The genus *Alternaria* belongs to the family Pleosporaceae, in the order Pleosporales, the largest of the class Dothideomycetes. A total of 275 *Alternaria* species were recognized according to the results of a study on *Alternaria* taxonomy based on morphological characteristics [6]. *Alternaria* is black-coloured due to melanin and forms fast-growing colonies. The surfaces of mature colonies may have a moist appearance due to the presence of numerous hyphae. The hyphae are septate and form conidiophores, also divided by septa. The conidia may be single or form chains. The *A. alternata* spores require dry air to be dispersed and relatively large, elongated and transversely septate [7]. Spores can separate from the conidiophore in dry air both passively and under the influence of strong winds. The similarity of the morphology and its variation according to the growth conditions makes it difficult the identification of the different *Alternaria* species [8].

## 2. Ecology of *Alternaria alternata*

Fungi can resist environmental stress, such as desiccation, producing spores as resting or dispersing forms. As the growth requirements of fungi are not very demanding, fungi are cosmopolitan organisms [9]. Fungal cells develop when environmental conditions, such as temperature and humidity, are advantageous for germination from spore or spore-like forms [10]. Spores require dry, warm, and windy weather to become airborne and to spread, so airborne spore counts reach maximum levels during sunny afternoons of late summer and early autumn and drop to zero during the winter [11].

*A. alternata* is a mesohygrophilic fungus. A study developed by Ren et al. showed that humidity conditions of 84–89 and 97% facilitated both fungal growth and mycotoxin production [12]. *A. alternata* tolerates a wide temperature range, from 2 to 32 °C, but its optimal temperature is around 20 °C. Pose et al. studied the influence of temperature on the germination and growth of *A. alternata* in a synthetic medium. The shortest germination time and the fastest growth rate were observed at 21–35 °C and 21 °C, respectively [13,14]. When the temperature drops below 15 °C, a reduction in the growth rate is observed. These mild temperature preferences allow *Alternaria* to grow in warmer climates, such as the Mediterranean area. Leon A. [15] and Bartra J. et al. [16] developed different studies in the region of Catalonia, Spain, and found that atmospheric variables can positively or negatively affect the release and permanence of fungal spores in the air and that the existence of correlation depends on whether the parameters are analysed in a punctual or cumulative manner. The authors observed a positive correlation between temperature and the release of conidiospores and a negative correlation between rainfall and conidiospores release. In a similar way, other fungi are influenced by relative air humidity. It has been observed that ascospores are positively correlated with high levels, as they require moisture for release, while conidiospores are negatively correlated with humidity, as conidiospores require dry conditions to disperse [6,15].

In tropical and subtropical areas, spore counts remain high throughout the year. In temperate climates, counts are highest from May to November and peak in late summer and autumn [17].

Kasprzyk et al. studied the presence of *Alternaria* spores in the air from April to September for 2 consecutive years in three different regions of Poland: Rzeszów, Lublin, and Poznań. Climate influences spore dissemination, but also other factors affect the spore airborne prevalence in some regions, such as the landscape and geobotanical conditions of the area. The authors found the highest spore counts in Poznań, the capital of the region with the highest urbanisation factor, which is surrounded by intensive agriculture practices [18]. Agricultural practices also influence environmental spore levels. During grain threshing, flax breaking, and thyme cleaning, the concentration of *Alternaria* fungi in the air is very high [19]. Other authors also found higher levels of fungal allergens in environments with high levels of bioaerosols, such as poultry farms and sawmills [20].

Although *Alternaria* spp. and *Cladosporium* spp. are predominantly outdoor allergens, exposure to their allergens can occur indoors [5]. Aerosol sampling studies showed that outdoor spore counts reached 7500 spores/m^3^ of air, while indoor spore counts could be as high as 280 spores/m^3^ [11].

Mould growth is facilitated by high humidity, often as a result of faulty insulation and outdated ventilation solutions, associated with low-income communities. The spores of the genera *Alternaria, Cladosporium, Aspergillus, Penicillium, Stachybotris* or *Ulocladium* can be found in carpets and bedding [21] and even in buildings with humidity problems or insufficient ventilation, which can lead to sick building syndrome [22,23], a set of symptoms and illnesses caused by indoor air pollution, in this case by allergens, volatile organic compounds or fungal mycotoxins [23]. In addition to moisture problems, houses with cockroach or cat infestations show a higher risk of *A. alternata* growing [24]. Furthermore, infections and large numbers of cats may be characteristic of homes without strict hygiene practices, so the concomitant lack of cleaning agents also encourages fungal growth, as well as animal droppings [23,25].

## 3. Allergy, Aerobiology and Prevalence of Sensitisation of *Alternaria alternata* and Other Fungi

Fungi cause a wide variety of clinical manifestations [5] as allergic rhinosinusitis [26], hypersensitivity pneumonitis or allergic alveolitis [27], oculomycosis [28], onychomycosis [29], skin infections [30], bronchopulmonary allergic mycosis [31,32] and anaphylaxis [33]. *Alternaria* is an opportunistic fungus that can cause alternariosis in immunocompromised patients, a particularly dangerous type of mycosis. However, the most severe allergic manifestation associated with fungi is severe asthma [4,11], which is characterised by decreased lung function and frequent exacerbations and can even lead to death. Emergency admissions due to asthma are associated with exposure to *Alternaria* and *Cladosporium* spores [34]. In addition, exposure to total spores, ascospores or basidiospores has been linked to the severity of allergic asthma [35,36].

Allergic sensitization necessarily involves previous exposition to the allergenic source in clinically relevant amounts [37]. The level of spores required to produce allergic symptoms in sensitised patients varies between fungal species.

The spore concentration needed to elicit allergic symptoms is 100 spores/m^3^ for *Alternaria* spp. [38], while for *Cladosporium* spp., the required spore levels are estimated to be 3000 spores/m^3^ [39]. However, not only spores but also fungal fragments contain detectable amounts of allergens and may function as aeroallergenic sources. These particles are present at considerably higher levels than spores and are of greater clinical importance in certain fungal species. In addition, fungal spores such as *A. alternata* release different amounts of the main allergen Alt a 1, depending on their development stage [40]. Therefore, traditional methods of identification and assessment of fungal exposure, based on morphological analysis and spore counts, are not sufficient to determine the levels of fungal allergens in the air. It has been found that allergic symptoms due to *Alternaria* spp. correlate with spore levels and Alt a 1 levels detected in the environment. However, the correlation between spore counts and Alt a 1 level is low [41]. Consequently, exposure to *Alternaria* spores has been described as the most important allergenic source associated with asthma in arid areas of the world [42,43].

The concentration of spores and fungal particles in the air in outdoor environments is usually very high throughout the year, exceeding 100–1000 times the concentration of pollen particles [44,45]. The highest spore levels are reached between spring and autumn. In fact, asthmatic patients sensitised to *A. alternata* have been reported to have more severe symptoms in late summer and early autumn [46], overlapping with the highest spore counts recorded in the air [47,48,49].

The levels of fungal spores in the air have a specific seasonal and daily cycle. These cycles depend on climate and weather conditions, circadian cycles, environmental factors or access of spores to substrates for their development [50]. A strong relationship has been observed between the occurrence of storm-associated asthma and sensitisation to *Alternaria* spp. Some factors that could explain the appearance of bronchial hyper-reactivity due to fungi of the genus *Alternaria* after a storm are a high release of spores into the air and their fragmentation into easily respirable particles, their transportation over long distances, a sudden decrease in temperature or an increase in ozone levels in the atmosphere [51]. Agricultural practices also contribute to increasing the levels of *Alternaria* spp. spores in the air, as it is a plant pathogen fungus which can grow on cereals [51]. Moreover, global climate change and CO_2_ concentration also appear to contribute to the stimulation of *Alternaria alternata* sporulation and total production of its antigens [52].

*Alternaria* can be found in indoor and outdoor environments, so exposure to fungal allergens can last for months, and their symptoms can be confused with symptoms produced by other allergens, such as pollens, mites and animal epithelia.

The actual prevalence of fungal sensitization is not well established, and the prevalence data tends to vary between authors and geographical areas of study. It has been estimated that between 3% and 10% of the world’s population is sensitised to fungi [37]. It has been suggested that in the Beijing area (China), allergy to fungi might be the most common aeroallergen [53]. A multicentre study carried out in 7 European cities by the Aerobiology Subcommittee of the European Academy of Allergology in 1997 concluded that 9.5% of the 877 patients studied with suspected respiratory allergy were sensitised to the fungi *A. alternata* or *Cladosporium herbarum*, with the highest prevalence in Spain, where 20% of patients were sensitised [54]. The European Community Respiratory Health Survey revealed that 4.4% of the general adult population (n = 11,355) were sensitised to *Alternaria* spp. [55]. Another study of 4962 patients with respiratory allergy showed a 19% of prevalence of sensitisation to species of the genera *Alternaria, Aspergillus, Candida, Cladosporium, Penicillium, Saccharomyces* and *Trichophyton*, determined by prick test [56,57]. In Poland, 102 out of 460 adult patients included in a study were found to be allergic to fungi, with the most frequent allergies being to *Alternaria* spp. (47.1%) and *Cladosporium* spp. (30.8%) [58]. A 2005 survey conducted by the Global Allergy Asthma European Network (GA^2^LEN) in 16 European countries revealed overall sensitisation rates to *A. alternata* and *C. herbarum* of 11.9% and 5.8%, respectively, with the highest prevalence in Northern Europe [59]. In 2009, in another large study conducted by GA^2^LEN, with the collaboration of 17 centres in 14 European countries, 273 of the 3,034 patients studied with suspected respiratory allergy (9%) showed sensitisation to *Alternaria* spp., with sensitisation rates ranging from 2% in Finland to 24% in Greece [60,61]. In Spain, in the 1995 Alergológica study carried out by the Spanish Society of Allergology and Clinical Immunology (SEAIC), 9.5% of the patients with suspected respiratory allergy included in the study were sensitised to *Alternaria* spp. and/or *Cladosporium* spp. [62]. More recent data has shown that the prevalence of allergy to fungi in Spain is around 3–10%, with *A. alternata*, *A. fumigatus*, *C. herbarum*, and *P. notatum* as the most prevalent species involved in sensitizations [63]. In the United States, the prevalence of *A. alternata* sensitisation in the general population (6–74 years of age) was 13% [64].

Among the main fungal genera causing allergy (*Alternaria, Aspergillus, Cladosporium* and *Penicillium*), the genera *Cladosporium*, in Northern Europe, and *Alternaria*, in the Mediterranean area, are the most frequently implicated fungi [54,65].

Sensitisation to a single fungal specie is very rare. In a study of 6000 patients in France, only 1% were monosensitised to *A. alternata* [66,67]. In Italy, 621 of the 2415 patients with allergic rhinitis included in the study (25.7%) were monosensitised to different allergens, and only 12 of these patients were found to be monosensitised to fungi (0.49% of the total) [68]. In the 2006 Iberian Study, which included 3225 patients with allergic rhinitis from Spain and Portugal, 399 patients were sensitised to *Alternaria* spp. However, of the 37% of patients who were monosensitised to different allergens, only 0.3% were sensitised to *Alternaria* spp. [69].

## 4. Allergen Distribution and *Alternaria alternata* Allergen Relevance

Allergens are antigens, usually non-pathogenic, that stimulate a hypersensitivity reaction. They are mostly proteins and can be found in different species of the Plantae, Animalia and Fungi kingdoms. Certain characteristics and biological functions of the antigens, such as low molecular weight, high glycosylation level, hydrolysis function, transportation or storage, appear to be related to allergenicity [70,71]. Enzyme-active proteins also may be related to the development of Th2 (IgE-mediated) immune responses [70,72] and influence the allergenicity of other allergens. It has been shown that fungi can cause asthma exacerbations because fungal allergens with enzymatic activity can break protein bonds between epithelial cells, enabling the accession of other allergens across epithelial barriers [73,74,75,76,77,78,79]. The enzymatic activity of allergens can also influence the allergenicity and antigenicity of extracts by modifying their protein and carbohydrate content [37].

The database of the World Health Organization and the International Union of Immunological Societies (WHO/IUIS, www.allergen.com, accessed on 1 December 2022) is the allergen database of reference in allergy, in which allergens have been included under restrictive criteria of inclusion [80]. The WHO/IUIS establishes the existence of 1090 allergenic proteins, 46.32% being allergens from organisms belonging to the kingdom Plantae, 42.64% to the kingdom Animalia, and 11.03% to the kingdom Fungi (Figure 1). A total of 120 allergens belonging to 31 species of fungi have been approved by the WHO/IUIS. Nevertheless, it is estimated that, among the whole number of fungal species described (more than 100,000), approximately 112 genera could act as allergenic sources [37].

Allergic reactions are mediated by IgE antibodies, mainly directed against protein epitopes. Nevertheless, it has been reported the existence of IgE antibodies with the ability to bind some carbohydrates is responsible for cross-reactivity (cross-reactive carbohydrate determinants; CCDs). The clinical relevance of CCDs is debated since cross-reactivity due to carbohydrates is not often associated with clinical symptoms. Indeed, they seem to be responsible for false positive results during the quantification of IgE antibodies in the serum of the patients [81,82]. β-glucans are the predominant carbohydrates in fungi (accounting for 50–60% of the dry weight of the cell wall) and show molecular sizes larger than 100 kDa [83,84,85,86,87]. These carbohydrates are not considered CCDs, but β-glucans have been reported to induce Th2 responses, such as in the case of the peanut allergen Ara h 1, which activates human monocyte-derived dendritic cells and the secretion of IL4 and IL13 [88,89]. Although some human studies suggest a dose-response relationship between exposure to fungal glucans and respiratory and skin symptoms and fatigue, little information is available on the allergenicity of fungal carbohydrates [90,91,92].

Fungi produce a wide variety of IgE-binding molecules, although not all of them are equally important. The importance of an allergen is determined by the prevalence of recognition among patients sensitised to the allergen source. Thus, an allergen can be classified as a major allergen when more than 50% of sensitised patients show allergen-specific IgE. Regarding the *A. alternata* allergens, 17 reactive IgE proteins have been identified and purified to date; 12 of them are accepted by WHO/IUIS and the other five are included in the Allergome database [5,93]. Alt a 1 is the main allergen [94,95,96,97,98] since more than 80% of *Alternaria*-allergic patients are sensitive to this allergen [95,99]. Alt a 1 is a 29 kDa glycoprotein composed of two subunits linked by disulphide bridges, with 16.4 and 15.3 kDa molecular weights [96]. Alt a 1’s biological function is unknown despite it being the most widely studied allergen of the specie. It is thought to be involved in spore germination, as it is predominantly located in the cell wall of the spores [100] and in plant infection processes [5,40,101,102].

The rest of the allergens described are considered minor allergens. However, there are studies showing the great relevance of some of these allergens, such as Alt a 13 (glutathione-S-transferase), Alt a 4 (protein disulphide isomerase) and Alt a 8 (mannitol dehydrogenase), which have shown IgE reactivity with sera from patients sensitised to the *Alternaria* genus in an 82%, 42% and 41% [100,103,104], respectively, or Alt a 70 kDa which, although not included in the WHO/IUIS database, it is seemed to produce positive skin test results in the 87% (14 out of 16) of the patients sensitised to *Alternaria* spp. [4,83,103,105,106,107]. The IgE reactivity observed for the other allergens described was lower than the observed for the Alt a 6 enolase, a thermostable protein considered a fungal panallergen, which showed IgE reactivity in 22% of *A. alternata* allergic patients [80,108,109,110,111]. Alt a 3 is a heat shock protein (HSP70) with a cell protective function against heat and oxidative stress, and Alt a 5 was identified as a P2 ribosomal acidic protein involved in the elongation of protein synthesis. These allergens showed IgE reactivity in 5% and 14%, respectively [112,113]. Alt a 12 is a P1 ribosomal acidic protein which, together with Alt a 5, forms dimers and is directly involved in protein processing [105]. Alt a 7, a homologue of the yeast protein YCP4, and Alt a 10, an NAD-dependent aldehyde dehydrogenase, showed IgE binding in 7% and 2%, respectively, in *Alternaria*-sensitised patients [105]. Alt a 14 is am Mn-dependent superoxide dismutase. Postigo et al. observed that 6.6% of *Alternaria* allergies could be attributed to reactions to MnSOD without sensitisation to Alt a 1. The authors also propose that together with Alt a 1 and enolase, the MnSOD should be included in the molecular set for the diagnosis of Pleosporaceae allergy [99,114].

The study of the sequence and conformational similarity of proteins allows the identification of homologous proteins in related or unrelated species. This could explain the existence of cross-reactivity between different allergen sources. Cross-reactivity between proteins may be due to the presence of common protein or carbohydrate sequences [37]. Examples of cross-reactive proteins include the enolases [110,115,116], manganese superoxide dismutases (MnSOD) [117], cyclophilins [118,119,120], glutathione-S-transferases (Alt to 13) [121], thioredoxins [122,123] and transaldolases [124]. In addition to these, cross-reactivity between serine proteases, ribosomal proteins, peroxisomal proteins and heat shock proteins has been described (Table 1). In Fungi, numerous Alt a 1 cross-reactive proteins have been described [37]. The comparison of the Alt a 15 sequence with other fungal serine proteases has revealed a sequence similarity of more than 50% in the genera *Curvularia* (Cur l 4), *Cladosporium* (Cla h 9, Cla c 9), *Penicillium* (Pen o 18, Pen ch 18), *Aspergillus* (Asp f 18) and *Rhizopus* (Rho m 2) [125]. Cross-reactivity has also been observed between other *A. alternata* allergens, such as Alt a 6 (enolase), Alt a 8 (mannitol dehydrogenase), Alt a 13 (glutathione-S-transferase), Alt a 14 (MnSOD) and Alt a NTF2, and *Aspergillus, Cladosporium, Penicillium* or *Curvularia* allergens [37,104,110,125,126,127,128]. Cross-reactivity between *A. alternata* allergens and food allergens has also been found, such as with edible mushrooms and spinach (*Alternaria*-spinach Syndrome) [129,130,131,132,133].

Given the high cross-reactivity and polysensitisation referred to *A. alternata* with other phylogenetically or non-phylogenetically related fungal species, several authors have suggested the need for a more specific diagnosis to other allergenic fungal sources after an initial diagnosis of sensitisation to *Alternaria alternata* [125]. This may be due to the different molecular compositions of the allergenic extracts depending on the strain, the methods used to obtain the allergens or even the origin of the sera used in the studies.

## 5. Diagnosis of *Alternaria alternata* Allergy and Its Immunotherapy

Allergic symptoms usually correlate to the presence of the allergenic source in the environment (pollen grains, mites and derivates, etc.). Regarding fungal allergy, patients who are allergic to *A. alternata* are frequently sensitized to other allergens such as grass and olive pollen, which are present in the atmosphere at the same time of the year, difficulting the diagnosis of *Alternaria* allergy. Traditional methods used to elaborate aerobiological calendars are based on pollen and spore detection. Nevertheless, it has been shown that *Alternaria* allergic symptoms do not correlate with the atmosphere spore counts, while a positive correlation has been observed between symptomatology and Alt a 1 levels [41]. Further studies relating symptoms and the detection of different allergens in the environment should be developed to improve the diagnosis of *Alternaria* allergy and help to predict the prognosis of the disease [134].

Allergen extracts are complex biological products used for the diagnosis and treatment of allergic diseases. They are obtained from the natural source (raw material) by extraction of its components. The biological composition of allergen extracts may change depending on the raw material origin, the time of collection and/or the production method, as well as the allergen elution methods and times, together with the purification processes used [135]. These factors will determine the final composition and quality of diagnostic products and vaccines and standardised products should be used as far as possible to ensure their efficacy and safety.

Since the 1990s, the development of recombinant allergen molecules has offered new diagnostic and therapeutic possibilities at a molecular level, allowing the identification of allergens to which a patient is sensitised, using purified natural or recombinant allergens in singleplex or multiplex measurement platforms [136,137,138]. Currently, the use of individual allergens for allergy testing in patients, known as Component Resolved Diagnosis (CRD), has improved allergy diagnostic accuracy and efficacy. The application of CRD has made it possible to minimise the problems of standardisation of allergen extracts, such as variability between fungal strains and between batches, the type of culture and technology used to prepare allergen extracts, as well as the stability of the extract once obtained. In recent decades, the application of molecular diagnostic procedures using individualised allergens has been helpful in the treatment of atopic individuals and allergic patients [139].

From the point of view of their composition, in most cases, allergy vaccines are safe, and their effectiveness is 80–90%, depending on the allergen or allergens involved. This efficacy has been proven in controlled studies in asthma and allergic rhinitis caused by mites, pollens and some fungi and animal epithelia [140]. However, the administration of allergenic extracts, whether subcutaneously or sublingually, is not exempt from risk. Generally, the most frequent reactions are local, with swelling or itching at the site of application, and although more severe reactions are rare, they occur within minutes of administration and can be life-threatening [141].

The efficacy of subcutaneous immunotherapy with allergenic extracts of *Alternaria* spp. and *Cladosporium* spp. has been demonstrated in adults and children; prospective, double-blind studies conducted so far have shown a symptomatological improvement in treated patients, as well as a decrease in IgE antibody levels and an increase in serum IgG levels [66,141,142,143,144,145,146,147,148]. However, fungal allergenic extracts have a complex composition of proteins, carbohydrates and other components that do not contribute to allergenicity but may produce adverse effects during treatment, so the safety of subcutaneous immunotherapy has been questioned [37]. It has been observed that treatment of asthmatic patients with extracts of *Alternaria* spp. and *Cladosporium* spp. produced a greater number of severe anaphylactic reactions than those produced by grass pollen, mites, or animal epithelia [149,150]. Thus, fungal extracts have been found to be less well tolerated than other allergenic extracts.

Different strategies have been developed to shorten allergy treatments and reduce the risk of developing adverse reactions. These are based on the modification of allergens or their adsorption to insoluble carriers (aluminium hydroxide, calcium phosphate, etc.). One of the most widely used methods, which has been shown to reduce allergenicity without compromising immunogenicity, is the use of glutaraldehyde (GA). GA is an aliphatic dialdehyde that binds to the free amino groups of amino acids such as lysine and arginine. Its addition to the allergenic extracts produces a chemical cross-linking between the proteins, obtaining allergen polymers or allergoids, which have altered immunological characteristics and a larger average molecular size much (100 to 1000 times larger) [151].

Another immunotherapy strategy is the isolation and characterisation of purified allergenic molecules from natural sources of allergens using classical biochemical methods [152,153]. However, single-allergen immunotherapy might not be successful in the total allergic population. Rodriguez et al. observed in 64 patients from a clinical trial that the percentage of recognition for Alt a 3, Alt a 4, and/or Alt a 6, Alt a 7, Alt a 8, Alt a 10 and/or Alt a 15 was 1.6%, 21.9%, 12.5%, 12.5%, and 12.5%, respectively, and 70.3% of the patients only recognized Alt a 1 [154]. In addition, these preparation methods are labour-intensive, and their efficiency is limited, making them less suitable for obtaining pure allergens in sufficient quantities for diagnostic and immunotherapy purposes [111,155].

## 6. Conclusions

*Alternaria alternata* belongs to the family Pleosporaceae and is one of the most prevalent fungi [6]. It is a cosmopolitan organism and can be found in both outdoor and indoor environments, leading to several clinical diseases. The most relevant diseases produced by *Alternaria* are allergic diseases. Fungi allergy has been largely confused with pollen and mite allergy, as symptoms, seasonality and distribution are similar to those sources. Despite the existence of more than 100,000 fungal species, the percentage of fungal allergens identified to date is poorly represented regarding the total of allergens identified in the Animalia and Plantae kingdoms. Fifteen *A. alternata* allergens have been identified to date, twelve of which are listed in the official database of the WHO/IUIS Allergen Nomenclature Subcommittee. Despite Alt a 1 being the *A. alternata* major allergen, other allergens, such as Alt a 6 or Alt a 14, have been suggested to be included in the diagnosis panel of fungal allergy. Given Alt a 1 is not the unique *Alternaria* allergen eliciting allergy symptoms, component-resolved diagnosis strategies should be applied to diagnose fungal allergy.

## Figures and Tables

**Figure 1 jof-09-00582-f001:**
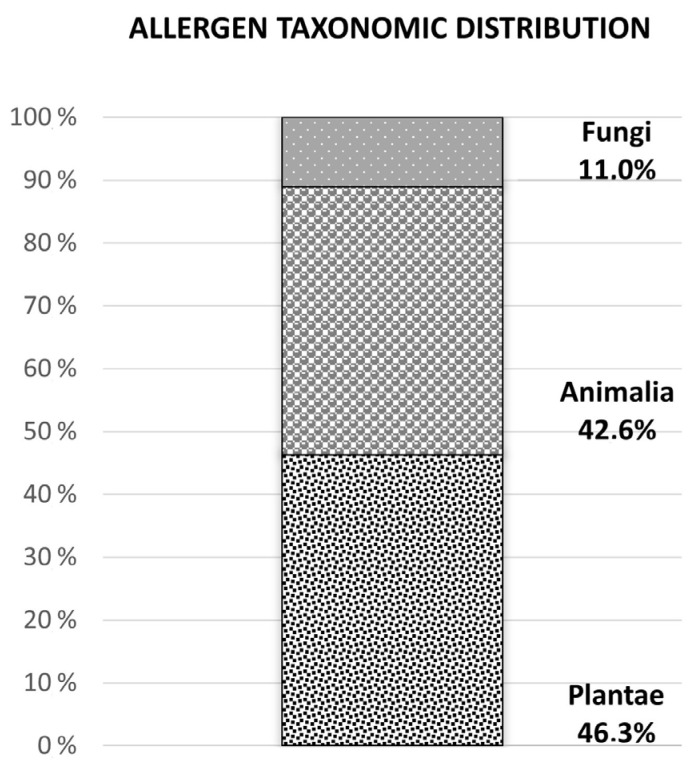
Taxonomic distribution of allergens described and accepted by the WHO/IUIS at the time of consultation (1 December 2022).

**Table 1 jof-09-00582-t001:** *Alternaria alternata* allergens and cross-reactivity with homologous proteins from other fungal species.

Allergen ^1^	Protein Type	MW (kDa)	Relevance	Potential Cross-Reactive Allergens (Structural Database of Allergenic Proteins E Score) IUIS/Allergen.org/Allergome.org Data Bases
Alt a 1	It has a unique, dimeric β-barrel structure that define a new protein family with unknown function found exclusively in fungi.	16.4 and 15.3 bands	Major allergen	Alt b 1 *Alternaria brasicola* and other 138 *Alternaria* species Ulo c 1 *Ulocladium chartarum* Emb a 1 *Embellisia allii* and other 6 *Embellisia* species Nim c 1 *Nimbya celosiae* and other 4 *Nimbya* species Sin fu 1 *Sinomyces fusoideus* Ste b 1 *Stemphylium botryosum* and other 2 *Stemphylium* species Ulo c 1 *Ulocladium chartarum* and other 11 *Ulocladium* species
Alt a 3	Heat shock protein 70	85	Minor allergen	Pen c 19 *Penicillium citrinum* (1.9 × 10^−27^) Mala s 10 *Malassezia sympodialis* (1.2 × 10^−3^)
Alt a 4	Disulfide isomerase	57	Minor allergen	
Alt a 5	Ribosomal protein P2	11	Minor allergen	Fus c 1 *Fusarium culmorum* (3.4 × 10^−27^) Cla h 5 *Cladosporium herbarum* (8.4 × 10^−25^) Asp f 8 *Aspergillus fumigatus* (2.0 × 10^−22^)
Alt a 6	Enolase	45	Minor allergen	Cla h 6 *Cladosporium herbarum* (1.2 × 10^−157^) Asp f 22 *Aspergillus fumigatus* (1.9 × 10^−154^) Pen c 22 *Penicillium citrinum* (4.7 × 10^−153^) Cur l 2 *Curvularia lunata* (8.4 × 10^−153^) Rho m 1 *Rhodotorula mucilaginosa* (2.5 × 10^−129^)
Alt a 7	Flavodoxin, YCP4 protein	22	Minor allergen	Cla h 7 *Cladosporium herbarum* (9.4 × 10^−61^)
Alt a 8	Mannitol dehydrogenase	29	Minor allergen	Cla h 8 *Cladosporium herbarum* (6.6 × 10^−91^)
Alt a10	Aldehyde dehydrogenase	53	Minor allergen	Cla h 10 *Cladosporium herbarum* (5.9 × 10^−168^)
Alt a 12	Aid ribosomal protein P1	11	Minor allergen	Cla h 12 *Cladosporium herbarum* (1.8 × 10^−30^) Pen cr 26 *Penicillium crustosum* (4.2 × 10^−28^) Pen b 26 *Penicillium brevicompactum* (6.8 × 10^−28^)
Alt a 13	Glutathione-transferase	26	Minor allergen	
Alt a 14	Manganese SO dismutase	24	Minor allergen	Asp f 6 *Aspergillus fumigatus* (5.5 × 10^−48^) Mala s 11 *Malassezia sympodialis* (4.9 × 10^−36^)
Alt a 15	Vacuolar serine protease	58	Minor allergen	Cur l 4 *Curvularia lunata* (4.3 × 10^−152^) Cla h 9 *Cladosporium herbarum* (5.3 × 10^−119^) Pen o 18 *Penicillium oxalicum* (2.4 × 10^−114^) Asp f 18 *Aspergillus fumigatus* (4.8 × 10^−111^) Pen ch 18 *Penicillium chrysogenum* (7.9 × 10^−111^) Cla c 9 *Cladosporium cladosporioides* (1.2 × 10^−99^) Rho m 2 *Rhodotorula mucilaginosa* (1.6 × 10^−73^) Tri r 2 *Trichophyton rubrum* (1.6 × 10^−37^) Pen ch 13 *Penicillium chrysogenum* (1.1 × 10^−23^) Asp v 13 *Aspergillus versicolor* (1.5 × 10^−20^) Asp f 13 *Aspergillus fumigatus* (2.8 × 10^−20^) Asp o 13 *Aspergillus oryzae* (9.8 × 10^−19^) Asp fl 13 *Aspergillus flavus* (9.8 × 10^−19^) Pen c 13 *Penicillium citrinum* (5.3 × 10^−15^)

^1^ World Health Organisation and International Union of Immunological Societies (WHO/IUIS) Subcommittee on Allergen Nomenclature, Structural Database of Allergenic Proteins (SDAP), International Union of Immunological Societies. Five other *A. alternata* allergenic proteins (Alt a TCTP, Alt a NTF2, Alt a 2, Alt a 9 and Alt a 70 kDa) are not included in this official allergen list but are already included in other allergomedical databases, such as the Allergome database (www.allergome.org (accessed on 1 December 2022)). From Sánchez P. et al., 2022 [97].

## Data Availability

Not applicable.

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
