# Peer review of "Going over Fungal Allergy: Alternaria alternata and Its Allergens"

_jof, 2023, doi:10.3390/jof9050582_

Round 1
Reviewer 1 Report
“Going over fungal allergy: Alternaria alternata and its allergens”. Journal of Fungi.
General revision:
I consider that the manuscript shows a review of the organisms that cause allergenic symptoms focusing on Alternaria as the primary allergenic fungi genera. However, the abstract does not show the manuscript's contents. I recommend rewriting the abstract according to the scheme of the manuscript. For example, the abstract shows the historical research of Alternaria but the manuscript does not comment on this.
In relation to the manuscript, the titles of the sections are general, waiting to read the general concepts of classification, ecology, aerobiology, and fungal allergens. However, the sections are focussed on Alternaria. I would change the title to would be focused to Alternaria.
Minor revisión:
Page 1. Lines 36-38. I would clarify conidia in the case of phylums Ascomycota and Basidiomycota, and sporangiospores in the case of phylum Zygomycota.
Page 1. Line 42: You must write the number with letters.
Page 2. Lines 41-52. You comment that the phylum Ascomycota is the most important allergenic source followed by Basidiomycota and Zigomycota. However, the genera that you show as examples are Ascomycota except for Malassezia which is Basidiomycota… I recommend rewriting these sentences.
Page 2. Line 52. Check Malassezia genera, it is misspelt.
Page 2. Line 61: What do you want to say with "A. alternata spores are dry"?
Page 2. Line 65. It is not clear if you are explaining the Ecology of general fungi or only Alternaria. The section describes the ecology of Alternaria, however, the title section is general
Page 2. Line 75. Alternaria is not a parasitic fungal of humans. In this case, it is an opportunist fungal.
Page 2, Line 88. Pose et al. need to be referenced.
Page 2, Lines 90-91. The symbol of degrees is incorrect.
Page 3, lines 91-101. Rua-Giraldo and Bartra el at. studied the conidia of Alternaria, not meiospores. In this paragraph, you cite mitospores, meiospores, conidiospores and ascospores… You could unify all these terms in conidiospores and ascospores… The reference to Leon is misspelt; the author is Rúa-Giraldo A.L.
Page 5. Lines 210-212- It needs references.
Page 5. Lines 223-224- I recommend eliminating this sentence. The range of the phylum used to describe which organisms provoke allergy is confusing… Cnydaria to Magnoliopsida… and what happens with fungal organisms?
Page 7. Lines 297-300: Why do you show here the results of Postigo et el. when they are referencing Alt a 1?
Page 9. Lines 374-398. These two paragraphs show different strategies to shorten allergy treatments… Have they been used Alternaria? What efficacy have they obtained?
Reviewer 2 Report
Many given descriptions discuss fungi and their properties in general, giving facts, generally famous. Especially – to the scientific community, which studies fungi and reads JoF. These descriptions should be at least very much compressed if not removed from the text.
Notes to the manuscript one by one:
Authors list: not clear, whose name letter M belongs to in the expression Mª José Martínez, if in the citatikon this name is expressed as Martínez J.
Abstract is poor and need adding of more data, which was aggregated as a result of this review. For example, ideas of conclusion are expressed better and may be used for the Abstract too.
Line 11: you wrote that sensitization to Alternaria is mainly seen in the Mediterranean area. But this is in contradiction with your own data, given in both manuscript by itself and in its conclusion, where you state that Alternaria is a world-important allergen. So, information should be adjusted.
Line 30-45: general description of fungi and their properties should be compressed with emphasizing just the data, which belongs to the matter of the manuscript, but does not repeats generally known facts on fungi.
Line 56-57 – mention number of Alternaria species here.
Lines 66-86 – give general and loose description about fungal properties again, sometimes it is not clear – were all fungi mentioned there or Alternaria only. Should be as much as it possible reduced just to the description to the Alternaria by itself or Alternaria in comparison with other fungi – to demonstrate analysis made in relation to this review, not just a description.
Lines 99-100. Its good to provide a description and/or pictures of morphological differences of ascospores and conidiospores of Alternaria if ascospores and conidiospores of this fungus are mentioned here. In case this description pertains to the fungi in general, it should be reconsidered toward reduction as mentioned above with a focus to Alternaria.
Line 110 – you mentioned region “with the highest urbanization factor and intensive agriculture practices”. But usually, urbanization is opposite to the intensive agriculture practices, so, maybe, these practices were seen around the city or in the neighboring area – needs clarification.
Lines 121-122: You wrote: “Fungi are also often found in damp buildings or with high humidity due to insufficient air conditioning or ventilation” – probably – “or with buildings with high humidity”…
Lines 122-128 – just a general description again, which does not clearly relate to the matter of the article, should be compressed.
Lines 144-145: The sentence: “The level of spores required to produce allergic symptoms in sensitised patients is unknown and varies between fungal species” contradicts with the next one, where these levels are given. So, you can write: The level of spores required to produce allergic symptoms in sensitised patients are known to little number of species and varies between them.
Line 162, Lines 178-184 – information, given here is repetitive, it was already mentioned in the Manuscript,
Lines 185-186: You wrote: “The actual prevalence of fungal sensitization is not well established and tends to vary 185 between authors and geographical areas of study…” However, sensitization is unlikely to vary between authors of study by themselves, ? this expression should be corrected.
Lines 189 and 205– the data of 1995 and 1997 is obsolete, I am sure it is possible to found a newer reference, given that diagnostic methods and opportunities are revolutionary changes since 1997.
Lines 210-212 – information is repetitive again.
Lines 222-237 – general description, should be reduced.
Lines 238-249 – the text hardly relates to fungi and disseminates a focus of the reader and manuscript in general, should be omitted or reduced much.
The same with lines 253-266, where CCDs are described. This information does not directly relate to the chosen topic.
Lines 301-307 – the phenomenon of cross-reactivity is generally understood by the professional community, its description should be either omitted or reduced just up to general mention.
Table 1 - as it appears from the description given in the text, the practical cross-reactivity of the mentuoined fungal allergens is unknown. So, it is better to write in the heading of the last column: allergens of the same biochemical class (potentially cross-reactive) or so
Line 340 – what allergomedical database is mentioned? Its name, website, authors/stakeholders?
Chapter 5 “Allergy diagnosis and immunotherapy” places description of the immunotherapy (treatment) before the description of possible diagnostic options and scientific bases of modern immunotherapy, which comprises separate molecules as a treatment for fungal allergy. It’s better to describe the problem consequently – possibilities of diagnostics and treatment then. Buy the way, Spain is the country, where almost the world's first molecular allergen immunotherapy (both subcutaneous and sublingual) for the Alternaria allergy was developed and is used in practice for the couple of years, this fact could be mentioned too.
Lines 358-361 – information is repetitive.
Conclusion is written well and this information be better used for the composition of the Abstract.
Reviewer 3 Report
Except for a few points this is quite a good "going over of fungal allergy" as per the title of the article. It is not especially different to other reviews but would make a good complement to the volume for which it has been written.
Reference 66 (Horst et al.) does not give the results of a survey on fungal allergy and polysensitisation. The authors It must be referring to another one-check and correct.
The sensitisation data concentrates on the Mediterranean and Iberian region. Fungal allergens are very important in regions where pollens are low such as the middle east and parts of China and Australia. Some reference to this would be appreciated.
I see that this overview has promoted the importance of fungal allergens from fourth to third compared to López Couso et al J. Fungi2021,7,631. https:// doi.org/10.3390/jof7080631. This is not a very important point but it would be best to emphasise its importance regionally and because of the severity of disease as well as the poor definition of allergens (especially for Cladosporium and Penicillin).
As well as the prevalence of detectably IgE binding or SPT responses it is very critical to consider the degree of sensitisation as can be most easily measured with IgE binding. This is critical for immunotherapy and for considering the importance of different components. For Aspergillus-allergic asthmatic patients there is a good concordance binding to Asp f 1 and binding to extracts with anti-Asp f 1 antibodies accounting for 30-100% of binding of patients (Crameri R, et al Automated specific IgE assay with recombinant allergens: evaluation of the recombinant Aspergillus fumigatus allergen I in the Pharmacia Cap System. Clin Exp Allergy 26(12):1411-9,1996) and similarly Alt a 1 binds IgE in 95% of allergic subjects at an average of about 15k IU/L representing 60% of the IgE binding value for the extract (Postigo I, Gutierrez-Rodriguez A, Fernandez J, Guisantes JA, Sunen E, Martinez J. Diagnostic value of Alt a 1, fungal enolase and manganese-dependent superoxide dismutase in the component-resolved diagnosis of allergy to Pleosporaceae. Clin Exp Allergy 41(3):443-51, 2011)
A major avenue of investigation has been the use of the purified Alt a 1 allergen for immunotherapy and how it affects the disease and responses to other Alternaria allergen components. I consider that a "going over" that does not analyse this is not satisfactory (Changes in the Sensitization Pattern to Alternaria Alternaria Allergens in Patients Treated with Alt a 1 Immunotherapy. Rodriguez D, et al. Journal of Fungi. 7(11), 2021). The reference also gives good data on the components recognised by allergic subjects.
The "going over" could provide precise information on the fungal allergen components (and lack of) now being used for component resolved diagnosis. At least some of the information can be found in Quan et al. Validation of a commercial allergen microarray platform for specific immunoglobulin E detection of respiratory and plant food allergens. Annals of Allergy, Asthma, & Immunology. 128(3):283-290.e4, 2022
With respect to undenominated allergens it could be noted that the sequence for the allergen Alt a 2 was that of a bacterial transposon (a molecular artefact) and Alt a 9 was only ever a band on a gel (and never had an IUIS designation). It is not known why Asp a 70K was never characterised except that later 2-D electrophoresis by the authors showed it consisted of several components and it is possible there weren't specificity tests. The authors never followed up their work moving to Alt a 1. Importantly for this paper the only data for allergenicity was published in reference 82. and references cited simply noted the single finding. Remove all the references except 82.
The idea that cats and cockroaches transport fungal spores into homes did not seem to be shown in my reading of reference 22 and I would need data to believe this could be a significant event.
Round 2
Reviewer 2 Report
It look like you either did not consider the most of my comments or did not provide appropriate version of the reviewed manuscript, where ALL changes - both all removed and added texts could be seen.
Or it looks like neither of required in my Review text shortages were not performed.
For example, it seems that the I can't compare previous and existing heading of the far right column of the Table 1. The only a new version is now seen here.
Also, there are some inappropriate headings appeared and incorrect expressions as well.
For example,
in Abstract you wrote: "Other four allergens are included in other medical databases", - specify these databases, please.
Chapter 1 is entitled: Classification, morphology and distribution of Alternaria alternata, but, as in the previous version, you write here about fungi in general and describe Alternaria then, so the title should be changed accordingly.
What is more, you wrote: “Fungi are eukaryotic, heterotrophic eukaryotic organisms and their reproduction can be sexual (spores) or asexual by fragmentation of the mycelium, budding (in yeasts), by conidia, as in the case of the phyla Ascomycota and Basidiomycota (non-flagellated asexual spores, produced in specialised hyphae) or sporangiospores, in the case of the phylum Zygomycota (produced in sporangia, cells normally supported by a foot).” This is very general information about fungi, which SHOULD BE KNOWN to the readers of JoF and other scientists, who are looking for the scientific information about fungi. In addition, what is written here is not correct. For example, fungi can reproduce both sexually and asexually by spores (e.g., ascospores, basidiospores, zygospores are products of sexual reproduction, while conidiospores and spores, created in sporangia are asexual or mitospores). In addition, fragmentation of the mycelium is referred as vegetative reproduction, not asexual. Sometimes budding and fission are referred as vegetative reproduction too. But, anyway, it is better to remove this paragraph at all. Nothing will change with a main idea of the text after that.
The same with Chapter 2. You haven't shortened general description of fungal ecology, so, paragraph contains a lot of general data about fungi, but the chapter is entitled: Ecology of Alternaria alternata, which is inappropriate again.
For example, this description: “Fungi feed by secreting enzymes directly into the environment and absorbing nutrients. When humidity and organic matter are abundant, fungal spores germinate and grow into filaments called hyphae, which later will develop into a mycelium. Fungi can also resist environmental stress, such as desiccation, producing spores as resting or dispersing forms. As growth requirements of fungi are not very demanding, fungi are cosmopolitan organisms (9) . They are widely spread in outdoor environments, although they can also be found in indoor environments where optimal conditions are generated as result of the activities of the building occupants. They contain melanin in the cell wall, which gives them protection against the germicidal effect of solar ultraviolet light and other types of radiation (9, 10) . Saprophytic species can be found on plants, soil, and foods (11)”, as it contains generally known data, which is NOT the matter of interest of the readers of the JoF as they mandatory should know general facts about fungi. So, these lines can be easily removed at all, so that the focus could be shifted to Alternaria indeed. And Manuscript and you lose nothing too.
The same is with this text: “Fungal cells develop when environmental conditions, such as temperature and humidity, are advantageous for germination from spore or spore-like forms. After invading the substrate by germination, they form irregular mycelial cells that resist the external environment and undergo a complex process of transformation (12). Spores require dry, warm, and windy weather to become airborne and to spread, so airborne spore counts reach maximum levels during sunny afternoons of late summer and early autumn, and drop to zero during the winter (13)”. It is extra too.
Again, if you entitled chapter 3 as “Allergy, aerobiology and prevalence of sensitisation of Alternaria alternata”, - so, please, write about Alternaria alternata, but not other fungi again!
Or give to this chaper following title: “Allergy, aerobiology and prevalence of sensitisation of Alternaria alternata in comparision with othr fungi” or so.
Chapter 4 is entitled: “Alternaria alternata allergens”. Are you sure that you describe these allergens only in this chapter? I am not, as description of plant and even animal allergens as well as other data on allergenicity of fungi is provided here.
using “Search” function it is easy to ensure that Alternaria is not mentioned in the lines 227-271 AT ALL (46 lines in total), which comprise more than 40 % of this Chapter volume. So, how this Chapter can be entitled as “Alternaria alternata allergens”?
And don’t forget to provide former titles of the Table 1 in general and its columns in particular if such were changed.
Chapter 5 is entitled: “Diagnosis of Alternaria alternata and immunotherapy”. But it is not clear, diagnosis of WHAT is mentioned here. Alternaria alternata can be detected rather than diagnosed in the environment and, obviously, this your chapter is not about its detection.
And, again, lines 338-364 do not mention Alternaria at all, which is almost a half of the Chapter volume. So, how it can be entitled with a focus on Alternaria?
So, please, these all issues should be harmonized before the Manuscript can be published.
Round 3
Reviewer 2 Report
Some work was done indeed, but I consider that in order to harmonize a text content with chapter titles,
Chapter 3 should be entitled "Allergy, aerobiology and prevalence of sensitisation to Alternaria alternata in comparison with other fungi" or so
and
Chapter 5 as "Diagnosis of Alternaria alternata allergy and its immunotherapy"
